# Therapeutic Prospection of Animal Venoms-Derived Antimicrobial Peptides against Infections by Multidrug-Resistant *Acinetobacter baumannii*: A Systematic Review of Pre-Clinical Studies

**DOI:** 10.3390/toxins15040268

**Published:** 2023-04-03

**Authors:** William Gustavo Lima, Maria Elena de Lima

**Affiliations:** Programa de Pós Graduação em Medicina-Biomedicina, Faculdade Santa Casa de Belo Horizonte, Belo Horizonte 30150-250, MG, Brazil

**Keywords:** antimicrobial peptides, *Acinetobacter baumannii*, antimicrobial development, arthropods, anura, venom, melittin, systemic infection, pneumonia, wound

## Abstract

Infections caused by multidrug-resistant *Acinetobacter baumannii* (MDR-Ab) have become a public health emergency. Due to the small therapeutic arsenal available to treat these infections, health agencies have highlighted the importance of developing new antimicrobials against MDR-Ab. In this context, antimicrobial peptides (AMPs) stand out, and animal venoms are a rich source of these compounds. Here, we aimed to summarize the current knowledge on the use of animal venom-derived AMPs in the treatment of MDR-Ab infections in vivo. A systematic review was performed according to the Preferred Reporting Items for Systematic Reviews and Meta-Analyses guidelines. The eight studies included in this review identified the antibacterial activity of eleven different AMPs against MDR-Ab. Most of the studied AMPs originated from arthropod venoms. In addition, all AMPs are positively charged and rich in lysine residues. In vivo assays showed that the use of these compounds reduces MDR-Ab-induced lethality and bacterial load in invasive (bacteremia and pneumonia) and superficial (wounds) infection models. Moreover, animal venom-derived AMPs have pleiotropic effects, such as pro-healing, anti-inflammatory, and antioxidant activities, that help treat infections. Animal venom-derived AMPs are a potential source of prototype molecules for the development of new therapeutic agents against MDR-Ab.

## 1. Introduction

*Acinetobacter baumannii* is a short (1.0–1.5 µm), Gram-negative coccobacilli characterized as strictly aerobic, catalase-positive, oxidase-negative, glucose non-fermenting, and nonmotile in physiological-biochemical assays [1]. Over the last decades, *A. baumannii* has globally emerged as a highly troublesome pathogen mainly because this bacterium shows a high capacity to become resistant to several antimicrobials such as betalactams, aminoglycosides, tetracyclines, quinolones, macrolides, and polymyxins [2,3]. Herein, hospitals in North America experienced a rise in resistance rates to carbapenems in *A. baumannii*, from 1.0% in 2003 to 58.0% in 2008 [4]. A recent meta-analysis showed that 13% of isolates of *A. baumannii* are resistant to polymyxins, the class of antimicrobials that is an ultimate choice to target such pathogens that have already become resistant to the frontline antibiotics such as β-lactams (including carbapenems), aminoglycosides, and fluoroquinolones [5]. Furthermore, multidrug-resistant *A. baumannii* (MDR-Ab) survives on inanimate surfaces of hospitals for up to one month due to its capacity to produce biofilms, causing contamination of patients [6]. This pathogen can cause a wide range of healthcare-associated infections (HAIs), including ventilator-associated pneumonia, septicemia, meningitis, osteomyelitis, peritonitis, endocarditis and wound, skin, soft tissue, urinary tract, ear, and eye infections [4,7,8]. The Centers for Disease Control and Prevention has reported that MDR-Ab is responsible for 12,000 infections annually only in the USA [9]. This context is particularly important because patients infected with MDR-Ab are more likely to die during hospitalization than those infected with susceptible strains of *A. baumannii* (91.7% vs. 48%; *p* = 0.01) [10]. Moreover, the global lethality of infections caused by MDR-Ab is 10.6%, with an estimated cost of $33,510 to $129,917 per infection [11].

Due to the great public health relevance associated with the dissemination of MDR-Ab, the World Health Organization (WHO) placed this pathogen in the first place of the priority list for research and development of active antibiotics published in 2017 [12]. In this context, antimicrobial peptides (AMP) stand out as promising sources of bioactive compounds against MDR-Ab [13,14,15]. AMPs are molecules produced by different living organisms as an innate defense mechanism [16]. These potential therapeutic agents have shown promising activity against several species of bacteria of medical interest (e.g., *Staphylococcus*, *Enterococcus*, *Streptococcus*, *Bacillus*, Enterobacteriales, *Pseudomonas*, *Acinetobacter*), including strains resistant to conventional antimicrobials such as methicillin-resistant *Staphylococcus aureus*, vancomycin-resistant *Enterococcus*, carbapenem-resistant Enterobacteriales, carbapenem-resistant *Acinetobacter baumannii*, and polymyxin-resistant *Pseudomonas aeruginosa* [17]. AMPs can act against pathogenic bacteria by direct mechanisms, mainly through interaction with the membrane (outer and inner) or inhibition of intracellular targets (metabolic pathways, enzymes, proteins synthesis, and structural molecules), and are also capable of modulating the antibacterial immune response mainly by stimulating the innate immune system through activation of toll-like receptors on defense cells [18]. Compared with conventional antimicrobials, AMPs have several advantages, such as a potent bactericidal effect, low potential to induce antimicrobial resistance, absence of residue formation in biological fluids, anti-biofilm effects, and synergistic activity when combined with bactericidal antimicrobials [13,14,15,16,17,18]. Therefore, AMPs can be considered a promising strategy to combat the spread of MDR-Ab infections worldwide.

Among the different sources of AMPs, animal toxins stand out due to this matrix’s great diversity of molecules [19,20]. In fact, AMPs with potent effects against MDR bacteria have been isolated from the venoms of several animals, such as ants, frogs, wasps, bees, spiders, scorpions, molluscs, snakes, and myriapods [20]. These molecules are often used as prototypes for developing analogues with better therapeutic properties, such as greater pharmacological potency, lower toxicity, and a more appropriate pharmacokinetic profile [21,22]. In this sense, for example, the addition of a polyethylene glycol (PEG) to a AMP derived from compounds of the venom of the spider *Lycosa erythrognatha* (called LyeTx I b) was able to reduce the toxicity and increase the stability of this molecule in vivo [23,24]. The peptides isolated or inspired by compounds recovered from animal toxins are also successfully used to treat several infections caused by MDR bacteria in preclinical models, including those induced by *A. baumannii* [25]. Therefore, this systematic review aims to summarize the state-of-the-art information related to AMPs that were originated or developed from molecules recovered from the venoms of animals in the treatment of MDR-Ab experimental infections.

## 2. Results

### 2.1. Study Selection

As shown in the PRISMA flowchart (Figure 1), 258 articles were identified by searching the selected databases—100 in Scopus (https://www.scopus.com/home.uri; accessed on 22 December 2022), 98 in PubMed/MEDLINE (https://pubmed.ncbi.nlm.nih.gov/; accessed on 22 December 2022), 32 in ScienceDirect (https://www.sciencedirect.com/; accessed on 22 December 2022), 22 in Web of Science (https://clarivate.com/webofsciencegroup/solutions/web-of-science/; accessed on 22 December 2022), and 6 in Biblioteca Virtual em Saúde (https://bvsms.saude.gov.br/; accessed on 22 December 2022). After excluding duplicates, 207 studies were further refined by ensuring that titles, abstracts, and keywords were in accordance with the inclusion criteria. The resulting 67 relevant studies were evaluated, and 60 were excluded based on the criteria described in Figure 1. The main exclusion moieties were studies that only included tests for anti-*A. baumannii* activity in vitro (n = 49), articles with peptides derived from animal fluids other than venoms (n = 4), and studies using in vivo infection models with other multidrug-resistant bacteria (n = 3). Afterwards, the reference list of the resulting 7 studies was checked, and a new article was included. Finally, 8 studies were used for critical reading and information extraction analyses [24,25,26,27,28,29,30,31]. The degree of agreement between the two authors was considered substantial, as shown by the kappa index (0.725).

### 2.2. In Vitro Biological Proprieties

The included studies identified the antibacterial activity against MDR-Ab of eleven different AMPs (Table 1). Most of the peptides were isolated or inspired from compounds recovered from arthropod toxin (9/11; 82%), including six molecules derived from the toxin of the honeybee (*Apis mellifera*; 6/11; 55%) [26,27,28,30], five from the moth hemolymph (*Hyalophora cecropia*; 6/11; 46%) [27,28], two from spider venom (*Lycosa erythrognatha*; 2/11; 18%) [23,32], and one from wasp venom (*Vespula lewisii*; 1/11; 9%) [31]. Only three peptides contain in their structure part of a vertebrate toxin combined with arthropod toxins or originated from a vertebrate toxin (3/11; 28%), both being developed from molecules recovered of anura (*Xenopus laevis* and *Hypsiboas albopunctatus*) [28,29]. Of the compounds included, only two (melittin and mastoparan) are isolated from honey bee and wasp venom (2/11; 18%) [26,30,31], and the others are obtained by synthesis performed from prototype compounds recovered from this biological matrix (9/11; 82%) [23,27,28,29,32]. Among the synthetic compounds, five (5/9; 56%) were developed as hybrid peptides by linking natural sequences of different peptides (Cecropin A-melittin analogues and K11) [27,28], and four (Hylin a1-11K, Hylin a1-15K, LyeTx I-b, and LyeTx I mnΔK) (4/9; 44%) were developed from complete synthesis, with some sequences of natural AMPs preserved [23,29,32]. Two studies used formulated peptides (2/11; 18%), with mastoparan incorporated into chitosan nanoparticles [31] and the LyeTx I-b peptide linked to polyethylene glycol (PEG) [23]. All other peptides were tested separately (9/11; 82%) [26,27,28,29,30,32].

Most studies evaluated the activity of AMPs against isolates of carbapenem-resistant *A. baumannii* (CRAB), which were categorized as extensively resistant (XDR) (6/8; 75%) [23,26,29,30,31,32]. The study by Rishi et al. [28] includes only isolates with a cephalosporin-resistant phenotype, while López-Rojas et al. [27] evaluated the antibacterial activity in colistin-resistant isolates, which is defined as pan-drug-resistant.

### 2.3. Physicochemical Proprieties

The physicochemical characteristics of the toxin-derived AMPs are summarized in Table 2. Among the eleven different peptides included, seven were considered short (less than 20 residues; 7/11, 63%) (CA(1–7)M(2–9), Oct-CA(1–7)M(2–9), CA(1–7)M(5–9), Hylin a1-11K, Hylin a1-15K, Mastoparan and LyeTx I mnΔK) [23,27,29,31] and four were categorized as medium-sized (between 20 and 50 residues; 4/11, 36%) (Melittin, CA(1–8)M(1–18), K11 and LyeTx I-b) [26,28,30,32]. None of the compounds were long-sized peptides (more than 50 residues). The molar mass ranged from 1478.91 Da for mastoparan [31] to 2847.45 Da for melittin [26,30]. The mean molar mass of the peptides included in this review was 2110.06 Da.

All peptides have basic amino acid residues and are positively charged at physiological pH (7.2). Interestingly, the AMPs active against MDR-Ab included in this review were rich in the amino acid lysine (K), and the amount of this residue of basic characteristic varied from eight for the hybrid peptide K11 [28] to three in melittin [26,30], mastoparan [31], and LyeTx I mnΔK [32]. Two other basic residues were also reported, but with a considerably lower frequency than lysine, in which two arginines (R) were identified in melittin [26,30], and histidine (H) was reported in only one residue in the hybrid peptide K11 [28]. The total amount of basic amino acid residues ranged from three in mastoparan [31] and LyeTx I mnΔK [32] to nine in the synthetic peptide K11 [28]. No acidic or cysteine amino acid residues were observed in the primary sequences of the AMPs, as shown in Table 2.

The electronic characteristic of the compounds was also studied. The net charge of AMPs ranged from +9 for K11 [28] to +4 for mastoparan [31] and LyeTx I mnΔK [32]. All compounds showed a high isoelectric point (pH 14), reinforcing the basic and cationic character of animal-toxin-derived AMPs active against MDR-Ab. With regard to water solubility, most peptides with anti-Ab-MDR activity were hydrophobic (7/11; 64%) [23,26,27,30,31,32], with only four molecules showing hydrophilic behavior (4/11; 36%) [27,28,29].

### 2.4. In Vivo Biological Proprieties

#### 2.4.1. Infection Models

All data related to in vivo experiments are summarized in Table 3. The antibacterial activity of animal-toxin-derived peptides was evaluated in different murine models of MDR-Ab infection. The studies used mainly non-neutropenic mice (6/8; 75%) [23,27,30,31,32], and only two studies used mice immunosuppressed with cyclophosphamide (150 mg/Kg for four days and one day after infection) (2/8; 25%) [26,28]. All studies used isogenic mice (Inbred) (8/8; 100%) [23,26,27,28,29,30,31,32], in which seven employed the BALB/c strain (7/8; 88%) [23,26,28,29,30,31,32] and only one used the C57BL/6 strain (1/8; 12%) [27]. Regarding the sex of the animals, half of the studies employed females (4/8; 50%) [23,27,28,32], three used males (38%) [26,29,30], and one of the studies did not specify the sex of the animals (1/8; 12%) [31].

Invasive infection models were most commonly employed (6/8; 75%) [23,26,27,29,31,32], with the therapeutic effect of AMPs being studied against intraperitoneal infection (4/8; 50%) [26,27,29,31] and pneumonia (2/8; 25%) [23,32]. Two studies used superficial infection models. Rishi et al. [28] evaluated the activity of the K11 peptide in infected open wounds, and Pashaei et al. [30] studied the activity of melittin in wounds induced by thermal burns. The size of the inoculum used varied considerably between the different models; however, it was in the range of 10^6^ CFU/mL to 10^8^ CFU/mL.

#### 2.4.2. Treatment Schemes

The treatment scheme varied according to the peptide employed and the infection model. In all cases, the therapeutic effect was evaluated after the establishment of the infection, and none of the studies evaluated the prophylactic activity of the compounds. All studies that used the intraperitoneal infection model employed the same route for treating the animals [26,27,29,31]. In this case, most authors used a single dose of peptides ranging from 1 mg/Kg for Hylin a1-11K and Hylin a1-15K [29] to 16 mg/Kg for Cecropin A-melittin analogues [27]. However, Askari et al. [26] used multiple doses of melittin to treat intraperitoneal MDR-Ab infection, employing four doses of 2.4 mg/Kg. After intraperitoneal infection, the studies waited 1 h to start the treatment [26,27,29,31], except for Rishi et al. [28], who waited 4 h post-infection.

In the model of pneumonia, Brito et al. [30] used two forms of the LyeTx I-b peptide (i.e., with and without PEG) intravenously in the treatment of lower airway infection, while Lima et al. [32] opted for local use, administering the LyeTx I mnΔK peptide intranasally. Both authors performed the treatment 2 h post-infection [23,32]. Moreover, the therapeutic management of wounds contaminated with MDR-Ab was also performed locally, in which Rishi et al. [28] incorporated the compound into a gel and applied it once a day for 21 consecutive days 4 h post-infection, and Pashaei et al. [30] used the peptide diluted in water only once 1 h post-infection.

#### 2.4.3. Lethality Assay

Two studies evaluated the lethality of MDR-Ab-infected animals after the treatment with AMPs [26,29]. Both studied the effect of the compounds in a model of peritoneal infection. Askari et al. [26] showed that four doses of melittin (2.4 mg/kg; i.p.) did not reduce the mortality of immunosuppressed mice infected with CRAB. However, Park et al. [29] showed that in non-neutropenic mice, the use of the peptides Hylin a1-11K and Hylin a1-15K in a single dose (1 and 2 mg/kg) reduced by 60% the lethality associated with systemic CRAB infection in the animals.

#### 2.4.4. Bacterial Load

The bacterial load was the efficacy indicator of the antimicrobial activity of AMPs most used in the studies, being evaluated by seven different works. Although melittin did not reduce the bacterial load in the peritoneal lavage and blood of mice with systemic infection [26], it decreased the microorganism count in wounds infected with CRAB and induced by thermal trauma [30]. On the other hand, all four cecropin A-melittin analogues tested by López-Rojas et al. [27] decreased the bacterial load in the peritoneal lavage of animals with systemic infection induced by a strain of polymyxin-resistant *A. baumannii*. Furthermore, the antibacterial effect of AMPs may also reduce the risk of bacteremia after peritoneal infection, as evidenced by the decrease in bacterial load in the blood observed in animals treated with mastoparan incorporated into chitosan nanostructures [31]. Interestingly, these same authors showed that free mastoparan does not exert any therapeutic effect, suggesting low stability of this peptide systemically [31].

Pulmonary bacterial burden was also significantly affected after the treatment with AMPs. According to Brito et al. [23], the use of intravenous pegylated LyeTx I-b peptide, but not its native form, decreased the bacterial load in the lungs of mice infected with a hypervirulent strain of CRAB. Similarly, Lima et al. [32] showed that a reduced version of the same compound, called LyeTx I mnΔK, also reduced the pulmonary bacterial load when used by inhalation.

In open wounds deliberately contaminated with cephalosporin-resistant *A. baumannii*, it was possible to conclude that using a gel formulated with the hybrid peptide K11 could accelerate bacterial clearance [28]. In this case, treated animals eliminate the bacteria in the wounds much faster than untreated animals.

#### 2.4.5. Physical Results

The study by Rishi et al. [28] shows that the use of animal-toxin-derived AMPs not only improves the microbiological aspects of superficial infections but also the recovery process of the animals. In addition to decreasing the bacterial load, the hybrid K11 peptide has healing effects, considerably accelerating skin re-epithelialization during the course of treatment. In addition, the body mass of the infected and untreated animals was lower than that of the animals that received the gel containing the K11 peptide. This indicates that the compound reduces the discomfort and pain generated by the wound, resulting in a higher intake of food by the animal, with consequent reduction in weight loss.

#### 2.4.6. Inflammatory and Oxidative Responses

One of the most important complications of the infectious process is the inflammation associated with the invasion of the pathogen in the tissue. Indeed, the loss of function and injury normally associated with the clinical manifestations of the infection is caused by the exacerbated immune response against the infectious agent. In the peritoneal infection model by CRAB, Park et al. [29] showed by histopathological analyses that the peptides inspired in compounds isolated from the *Hypsiboas albopunctatus* toxin (Hylin a1-11K and Hylin a1-15K) inhibited the inflammation associated with infection not only at the site of action of the drug (peritoneal cavity), but also in abdominal organs. In this context, a reduction in inflammatory foci was observed in the kidneys, liver, lungs, and spleen of treated animals. This indicates that, in addition to potent antibacterial action, AMPs can also modulate the immune response by reducing the deleterious effects of inflammation.

Another component of the innate immune system that can cause tissue damage after the establishment of an infectious focus is reactive oxygen species (ROS). These highly reactive agents can cause peroxidation of lipids in cell membranes and react with proteins, destabilizing them and causing effects that lead to tissue lesion. Interestingly, Rishi et al. [28] verified that the peptides reduce the lesions induced by ROS in open wounds infected with MDR-Ab, and that part of this effect can be associated with the ability of the peptide to stimulate the activity of the natural antioxidant system as the enzyme catalase.

## 3. Discussion

The rapid spread of HAIs by MDR-Ab has pressured research and development centers worldwide to search for new bioactive compounds effective against this pathogen [4,5,12]. In this context, AMPs stand out as potential prototypes for developing antimicrobials with activity against this superbug, thus helping to expand the narrow therapeutic arsenal currently available for the treatment of MDR-Ab infection [33]. AMPs are particularly frequent in animal venoms. However, although many studies have evaluated the activity of animal-venom-derived AMPs in vitro, little effort has been made to characterize their therapeutic potential in rodent models of MDR-Ab infection [34]. Furthermore, no study has summarized these results to highlight the main challenges related to the use of AMPs against MDR-Ab-induced infections in clinical practice [35]. Therefore, this study aims to critically evaluate, through a systematic review following the PRISMA 2021 standards [36], the results regarding the use of animal-venom-derived AMPs in the treatment of experimental infection by MDR-Ab.

AMPs can be derived from several sources such as plants, bacteria, fungi, and animals. In this review, where the main objective was to evaluate the AMPs isolated or inspired by compounds from animal venoms, we observed that within this specific source, arthropod venom was highlighted as the main origin of active compounds against MDR-Ab. The higher frequency of studies with arthropods is justified by their remarkable biodiversity and the medical relevance of accidents with these organisms (especially bees, wasps, spiders, and scorpions) [37,38]. Most studies used peptides inspired by melittin, an important AMP isolated from European bee (*Apis mellifera*) venom. This compound is currently available for clinical use in topical formulations for cosmetic purposes. The cytolytic action of melittin has been known since the 1950s, and this peptide forms pores in the membrane, inducing the lysis of pathogenic bacteria [39,40]. Melittin is often used as a model for the development of new bioactive molecules against multidrug-resistant bacteria, including MDR-Ab, as evidenced in this study, since it is well-known, clinically available, has a high bacteriolytic effect, and shows easy availability through bee venom purification or biotechnological strategies [34,39,40,41].

Due to its great genetic plasticity, *A. baumannii* can acquire resistance genes against virtually all classes of antimicrobials [1,8]. Epidemiologically, however, of particular concern are carbapenem-resistant strains. The carbapenems, represented by ertapenem, imipenem, doripenem, and meropenem, act by inhibiting cell wall transpeptidation [42]. However, the production of enzymes that hydrolyze the beta-lactam ring of carbapenems, known as carbapenemases, considerably reduces the sensitivity of *A. baumannii* to these drugs [4,43]. The World Health Organization (WHO) estimated a global rate of resistance to carbapenems in *A. baumannii* of 63.2%, with some countries, such as Greece, reporting 90–100% resistance [44,45,46]. In a recent meta-analysis that included 1226 *A. baumannii* isolates, 53.2% were positive for oxacillinase, a carbapenemase belonging to Ambler’s D class known to be the most frequent in this bacterium [4]. Furthermore, in the US, the lethality of CRAB is higher than in other non-fermenting multidrug-resistant Gram-negative bacteria, such as carbapenem-resistant *Pseudomonas aeruginosa*. Of the 1048 cases of CRAB reported in military hospitals in the country, 30.3% evolved to death, and despite a higher number of infections (8204), CRPA was responsible for a lethality rate of 24.5% [45]. Therefore, the high incidence of resistance to carbapenems in *A. baumannii* justifies the greater emphasis given to this specimen in the included studies [46].

All AMPs active against MDR-Ab are positively charged at physiological pH. This physical-chemical property is relevant because it guarantees the pharmacological activity of these molecules. The outer membrane of *A. baumannii* is rich in phosphorylated lipids and negatively charged proteins, causing its surface to have an excess of electrons that gives it an overall negative charge [1]. Thus, when AMPs are positively charged, they can electrostatically interact with the membrane of this pathogen and position themselves adequately to attack this biolayer. After the initial interaction, AMPs typically insert between the phospholipids of the membrane, producing pores that destabilize this structure and lead to cell lysis [34,47,48,49]. In addition, due to their positive charge, these molecules displace divalent cations bound to the plasma membrane, such as Ca^2+^, Mg^2+^, and Zn^2+^, which are essential cofactors for the metabolic activity of the bacterial cell [49]. Therefore, the positive charge of peptides is central to their action against MDR-Ab.

Another relevant feature is that all animal-venom-derived peptides active against MDR-Ab are rich in basic amino acid residues. This characteristic is directly related to the overall charge of these structures since these residues, usually formed with free amine groups, are protonated at physiological pH, giving the compound a positive charge [47,48,49]. Furthermore, basic residues often interact with acidic regions in intracellular molecular targets, which may contribute to the antimicrobial activity of these compounds [37]. Indeed, many enzymes involved in microbial energy metabolism are rich in residues of an acidic character because the pH inside the bacterial cell is lower (pH 5.8–6.3) than the pH of mammalian cells (7.2) [50,51]. Additionally, the basic residues of the AMPs, when in contact with the acidic interior of the bacteria, become protonated, making it difficult for these compounds to leave the bacterial cell since the plasmatic membrane is not very permeable to charged molecules. This “collection” of molecules with a positive charge inside the cell can act on different intracellular targets by ionic interactions (e.g., cation-π, cation-anion), causing their inhibition or even interact with the inner face of the bacterial cell membrane, enhancing the damage in this cellular envelope [47,48,49].

Among basic amino acid residues, it is important to highlight that all bioactive peptides are rich in lysine. Interestingly, it has been shown that the use of this amino acid potentiates the activity of antimicrobial substances against *A. baumannii*. Deng et al. [52] revealed that the use of L-lysine (40 nM) significantly increases the bactericidal activity of gentamycin, kanamycin, and amikacin against CRAB. A similar effect was observed with the Gram-negative bacteria *Escherichia coli* and *Klebsiella pneumoniae* and the Gram-positive *Mycobacterium smegmatis*. Lysine stimulates the proton-motive force (PMF) that contributes to the entry of antimicrobial substances into bacterial cells [52]. This activation occurs because, due to its basic characteristic, lysine induces a pH variation in the bacterial cell that activates the PMF, stimulating the internalization of antimicrobial substances. Indeed, treating cells with magnesium sulfate, which inhibits pH changes in the bacterial cell, blocks the effect of lysine on the activity of aminoglycosides [52,53]. Thus, the presence of lysine in these AMPs may contribute to the entry of compounds into the bacterial cell and potentiates their biological activity. Furthermore, lysine is known to stimulate the generation of reactive oxygen species (ROS) in bacterial cells [52]. This propriety may be an additional antibacterial mechanism of AMPs since ROS accumulation within bacteria can trigger cell death or arrest cell growth by damaging proteins, DNA, RNA, and lipids of the membrane [54,55,56].

The MDR-Ab infection models used mainly inbred mice from the BALB/c strain. The BALB/c mouse is among the most widely used models in biomedical research and is particularly employed in immunology and infectious disease studies. It is because these animals have a less pronounced Th_1_ response and are, therefore, more susceptible to infectious diseases. On the other hand, with BALB/c mice, Th_2_ cells are easily triggered by immunization, meaning that this animal strain is an exceptional responder to immunization [57,58]. *A. baumannii* is known to cause bacteremia and infections of wounds in the respiratory, gastrointestinal, and genitourinary tracts [1,4] and has increasingly become a cause of bloodstream infections and pneumonia [7]. Corroborating the incidence profile of this bacterium, most studies used models of systemic infection, followed by pneumonia and wound infection. Therefore, the evaluation of the antibacterial effect of AMPs against *A. baumannii* covers the main spectrum of diseases that this bacterium causes.

Treatment schemes varied considerably between the models evaluated, but the start of peptide use was instituted between 1 h and 4 h post-infection. In a model of CRAB-induced pneumonia, Bergamini et al. [59] showed that the intranasal instillation with 50 µL of a bacterial suspension at 10^8^ CFU/mL resulted in a maximum lung bacterial load in the animals after 2 h (5.80 Log_10_ CFU/lungs), in which it remained approximately constant for up to 26 h (5.10 Log_10_ CFU/lungs) when it begins to decline. In turn, in a model of septicemia induced by the intraperitoneal administration (10^7^ CFU per animal) of a hypervirulent strain of *A. baumannii*, it was observed that after 4 h, the bacterial load is maximum in the peritoneal cavity (5.5 log_10_CFU/mL), blood (5.2 log_10_CFU/mL), lung (4.3 log_10_CFU/mL), spleen (4.2 log_10_CFU/mL), and kidney (3.9 log_10_CFU/mL). In this study, the amount of bacteria in these organs significantly reduced after 24 h and is practically undetectable after 48 h, except for the lung, where they remain at low cell density up to 168 h post-infection [60]. Therefore, in the in vivo assays that sought to verify the activity of animal-venom-derived AMPs, the authors used each of the models at the peak of the infection, where the bacterial load is highest, and the cells are in logarithmic growth.

The most important variables to evaluate antibacterial efficacy are lethality and bacterial load. For a compound to be considered a promising antibacterial, it must protect against the lethality induced by the infectious agent and must be able to reduce the microbial load since this parameter is directly proportional to the risk of death caused by the infection [61]. In invasive infection models (systemic infection and pneumonia), all tested AMPs reduced the mortality and bacterial load of MDR-Ab, except for melittin, which was unable to improve these factors after intraperitoneal administration in a systemic infection model [26]. This activity profile can be justified by the toxicity of this AMP, which has a high hemolytic potential (HC_50_ 0.44 µg/mL) and low toxic dose in vivo (LD_50_ of 4.95 mg/kg i.p. and sub-lethal dose of 2.4 mg/kg) [26]. Corroborating this finding, the use of hybrid peptides containing parts of the melittin sequence, which are less toxic [27,28], and the use of melittin topically to treat MDR-Ab wounds [30] showed promising in vivo results.

Moreover, AMPs have additional therapeutic effects that may aid in the post-infection healing process. Rishi et al. [28] showed that the use of the AMP K11 not only reduced the bacterial load in an MDR-Ab-induced wound model but also stimulated re-epithelialization and reduced oxidative damage and weight loss in animals, accelerating complete recovery when compared with animals treated only with the positive control (meropenem). This occurs because, unlike conventional antimicrobials that only have an antibacterial effect, AMPs are versatile molecules that stimulate other biochemical and physiological processes that help repair tissue damage caused by the infectious process [62,63]. Park et al. [29], in turn, showed that using hybrid AMPs reduced the inflammatory infiltrate in the kidneys, liver, lungs, and spleen of mice infected with MDR-Ab intraperitoneally, suggesting that these compounds are also important inhibitors of the inflammatory process induced by pathogens. Corroborating these findings, some studies show that AMPs can readily penetrate the cell wall barrier and neutralize the LPS of bacterial cells. This effect is associated with the ability of AMPs to combine with LPS competitively and inhibit the transport of LPS and inhibit the LPS binding to lipopolysaccharide-binding protein in immune cells. However, AMPs can also inhibit the production of biological cytokines, which engineer the inflammatory process [34,63,64].

This systematic review has some limitations. First, multiple variables influence the overall therapeutic effect of the studied compounds, such as the wide variety of animals, the scheme of treatment used (dose, period, administration via), and the AMP employed. Second, studies have neglected evaluating the anti-MDR-Ab activity of the peptides obtained from venoms of different animals, such as reds, sea sponges, scorpions, snakes, fish, salamanders, and lizards, which limited this review to peptides derived from the venom of a small group of animals, mainly representing some species of the arthropods and anura. Moreover, the activity of animal-venom-derived peptides against polymyxin-resistant *A. baumannii*, a pan-drug resistant strain that is not susceptible to all antimicrobials currently available in the clinics, has been explored by only one study. Third, no study has evaluated the possible mechanisms of the antibacterial action of peptides against MDR-Ab. Fourth, the limited number of included studies reveals that the evaluation of the antibacterial activity of animal-venom-derived AMPs is largely neglected, thus limiting the knowledge of the real therapeutic potential of these peptides. In addition, none of the included studies assessed the pharmacokinetic properties or stability of the peptides in a biological context, which are directly related to assessing potential pharmacological compounds. Fifth, studies neglect to assess the toxicity of AMPs in vivo, which limits the ability to measure the real therapeutic role of these agents. In this sense, knowing how the compounds act in the different systems and verifying, through biochemical dosages, the renal, hepatic, cardiac, and pulmonary function, after the administration of these antimicrobial agents, is essential to verify the potential for clinical use of AMPs in the future. Sixth, none of the studies identified the limitations of the experimental design employed, which may compromise the observed correlations. Finally, systematic reviews do not imply a causal relationship, and the outcomes are always subject to the influence of different biases of the included studies.

## 4. Materials and Methods

The antibacterial activity of animal-toxin-derived peptides against MDR-Ab was studied by a systematic review performed according to the principles described in the Cochrane Handbook [65]. The steps related to the search, selection, extraction, analyses, and interpretation of data of interest were conducted according to the Preferred Reporting Items for Systematic Reviews and Meta-Analyses (PRISMA) 2021 statement [36].

### 4.1. Search Strategy

A systematic search was performed in 5 databases (i.e., PubMed/MEDLINE, Scopus, Web of Science, ScienceDirect, and Biblioteca Virtual em Saúde). To determine the descriptors used in the search strategies, the Medical Subheading Terms (MesH) were employed to define the keywords in English, and the Descritores Virtual em Saúde (DeCS) to define the keywords in Portuguese and Spanish. The term “antimicrobial peptides” was combined using the boolean AND with the terms “venom” OR “toxin”. In all combinations, the term “acinetobacter baumannii” was also used, as in the example: “antimicrobial peptide” AND “venom” AND “toxin”. The detailed search strategy used is shown in Appendix A. In addition, the reference lists of all the included articles and relevant narrative reviews in this field were evaluated for any relevant articles. Using Scopus, we also checked the authors who cited the included studies to identify other works eligible for this systematic review.

The search was conducted until 24 December 2022. We have restricted the search to the studies written in English, Portuguese, and Spanish, but no date limit was established.

### 4.2. Inclusion and Exclusion Criteria

In order to identify studies of interest, we applied the PICOS strategy [66] as follows: Population—rodents (mice, rats, hamsters, or rabbits) infected with MDR-Ab; Intervention—treatment with animal-toxin-derived antimicrobial peptide; Control—treatment with placebo (saline or PBS); Outcomes—Primary: Mortality and Bacterial load; Secondary: Clinical signals and inflammation process (qualitative analysis); Study design—in vivo studies conducted in rodents (pre-clinical assays).

The exclusion criteria were: (i) review articles (narrative, systematic, and meta-analysis), notes, correspondences, editorials, letters to the editor, and abstracts; (ii) studies characterizing only the in vitro antibacterial activity of animal-toxin-derived peptides against MDR-Ab; (iii) studies that do not identify the resistance profile of the *A. baumannii* isolates employed or that include only isolates sensitive to conventional therapies; (iv) studies in which none of the main outcomes (mortality and bacterial load) has been the subject of study; (v) studies that do not describe the chemical characteristics of the peptides used or that do not employ molecules isolated or derived (inspired) from molecules recovered of animal venoms; and (vi) studies that do not describe the origin of the peptides used. In cases where the article was in accordance with inclusion criteria, but the full text was unavailable, the corresponding author was contacted by e-mail three times (with 14-day intervals between them), and the articles included if they were received by the last point of contact [67].

### 4.3. Selection of Studies

In the first phase of the selection, two independent researchers searched the databases. Duplicate records were deleted using Rayyan^®^ (Version 2022; Doha, Qatar, 2022) software. Next, the titles, keywords, and abstracts of the studies were screened following the PICOS eligibility criteria. Thereby, studies that evaluated the therapeutic effects of animal-toxin-derived peptides against MDR-Ab using in vivo models were selected and then evaluated by full-text review. Any discrepancies were resolved by discussion between the researchers until reaching a consensus. After a full analytical reading, all data of interest were summarized in a table for further critical analysis and interpretation.

The kappa concordance index was determined to define the concordance rate between the researchers. The agreement between evaluators is optimal when the kappa remains between 0.8 and 1.0; very good to kappa between 0.7 and 0.8; good to kappa between 0.5 and 0.7; moderate to kappa between 0.3 and 0.5; and low when the kappa index is less than 0.3 [68]. If agreement is categorized as low, the search is performed again.

## Figures and Tables

**Figure 1 toxins-15-00268-f001:**
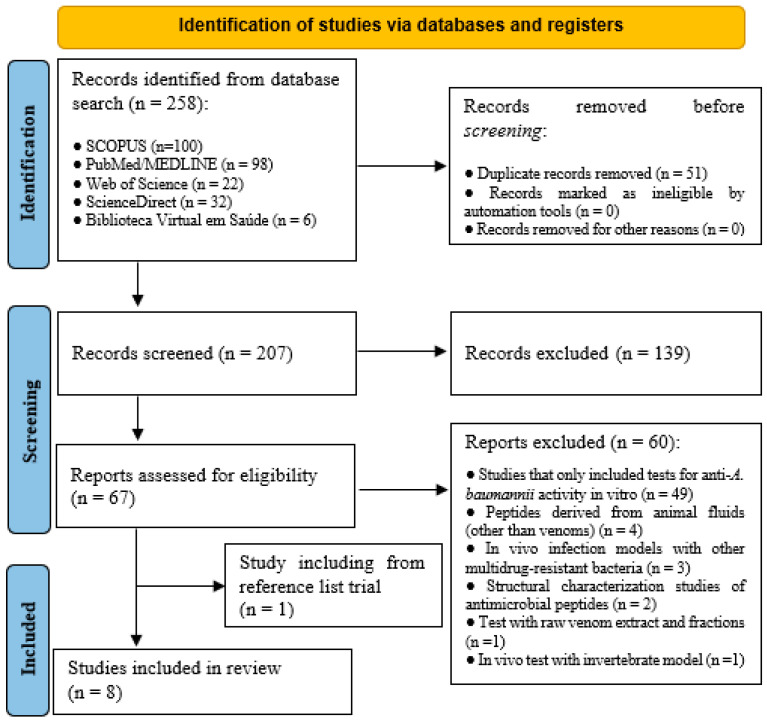
Flowchart of the selected articles for the systematic review according to the PRISMA criteria.

**Table 1 toxins-15-00268-t001:** In vitro biological proprieties of toxin-derived antimicrobial peptides active against multidrug-resistant *Acinetobacter baumannii*.

Peptide		Resistance of Tested Isolates	Reference
Name	Specie and Chemical Origin		
Melittin	Apis mellifera Natural	Carbapenem (XDR)	Askari et al., 2021 [26]
CecropinA/melittin analogues-(four compounds)	Hemolymph of Hyalophora cecropia (Cecropin A) and Apis mellifera (Melittin) Hybrid	Colistin (PDR)	López-Rojas et al., 2011 [27]
K11	Hemolymph of Hyalophora cecropia (Cecropin A); Apis mellifera (melittin), and Xenopus laevis Hybrid	Cephalosporins (MDR)	Rishi et al., 2018 [28]
Hylin a1-11K and Hylin a1-15K and	Hypsiboas albopunctatus Venom-derived peptide	Carbapenem (XDR)	Park et al., 2022 [29]
Melittin	Apis mellifera Natural	Carbapenem (XDR)	Pashaei et al., 2019 [30]
Mastoparan	Vespula lewisii Natural	Carbapenem (XDR)	Hassan et al., 2021 [31]
LyeTx I-b	Lycosa erythrognatha Venom-derived peptide	Carbapenem (XDR)	Brito et al., 2021 [23]
LyeTx I mnΔK	Lycosa erythrognatha Venom-derived peptide	Carbapenem (XDR)	Lima et al., 2021 [32]

MDR: multidrug-resistant bacteria; XDR: Extensively drug-resistant bacteria; PDR: Pan-drug-resistant bacteria.

**Table 2 toxins-15-00268-t002:** Physicochemical proprieties of toxins-derived antimicrobial peptides active against multidrug-resistant *Acinetobacter baumannii*.

Name	Reference	Sequence	Residues Number	Molecular Mass (Da)	Net Charge at pH 7 ^A^	Polarity ^B^	Amino Acids Residues		
							Basic	Acid	Cys
Melittin	[26,30]	**GIGAVLKVLTTGLPALISWIKRKRQQ**	26	2847.45	Positive (5+)	Hydrophobic	5	0	0
Cecropin A-melittin analogues		CA(1–8)M(1–18)—**KWKLFKKIGIGAVLKVLTTGLPALIS-NH_2_**	26	2793.78	Positive (6+)	Hydrophobic	5	0	0
	[27]	CA(1–7)M(2–9)—**KWKLFKKIGAVLKVL-NH_2_**	15	1770.19	Positive (6+)	Hydrophilic	5	0	0
		Oct-CA(1–7)M(2–9)—**octyl-KWKLFKKIGAVLKVL-NH_2_**	15	1898.50	Positive (6+)	Hydrophobic	5	0	0
		CA(1–7)M(5–9)—**KWKLFKKVLKVL-NH_2_**	12	1544.07	Positive (6+)	Hydrophilic	5	0	0
K11	[28]	**KWKSFIKKLTKKFLHSAKKF-NH_2_**	20	2493.09	Positive (9+)	Hydrophilic	9	0	0
Hylin a1 analogues	[29]	Hylin a1-11K—**IAKAILPLALKALKNLIK-NH_2_**	18	1930.51	Positive (5+)	Hydrophobic	4	0	0
		Hylin a1-15K—**IAKAILPLALKALKKLIK-NH_2_**	18	1944.58	Positive (6+)	Hydrophilic	5	0	0
Mastoparan	[31]	**INLKALAALAKKIL-NH_2_**	14	1478.91	Positive (4+)	Hydrophobic	3	0	0
LyeTx I-b	[23]	**IWLTALKFLGKNLGKLAKQQLAKL-NH_2_**	24	2695.34	Positive (6+)	Hydrophobic	5	0	0
LyeTx I mnΔK	[32]	**IWLTALKFLGKNLGKL-NH_2_**	16	1814.26	Positive (4+)	Hydrophobic	3	0	0

Blue letters: Basic residues; Red letters: Acid residues; Yellow letters: Cysteine. ^A^ The isoelectric point of all peptides studied was pH 14. ^B^ The polarity was defined by the coefficient of solubility in water at 25 °C. Peptides with low water solubility (i.e., solubility coefficient lower than 1 g/L) were considered HYDROPHOBIC. Peptides with high solubility in water (i.e., solubility coefficient greater than or equal to 1 g/L) were considered HYDROPHILIC.

**Table 3 toxins-15-00268-t003:** In vivo biological proprieties of toxin-derived antimicrobial peptides active against multidrug-resistant *Acinetobacter baumannii*.

In Vivo Model (Mice Lineage, Infection Type and Infection Protocol)	Treatment (Molecule, Dose and Therapeutic Regime)	Main Outcomes Associated with the Treated Group	Reference
Neutropenic BALB/c mice (male) ^a^ Peritoneal infection 10^7^ CFU/mouse dissolved in 500 µL PBS i.p.	Melittin 2.4 mg/kg 1 h post-infection and repeated every 12 h up to 36 h (four injections) (i.p.; diluted in 500 µL of PBS)	- No decrease in the bacterial loads in peritoneal fluid - No decrease in animal mortality - No decrease in the positivity of blood culture	Askari et al., 2021 [26]
Non-neutropenic C57BL/6 mice (female) Peritoneal infection 10^6^ CFU/mouse dissolved in 250 µL of saline i.p.	CA(1–8)M(1–18), CA(1–7)M(2–9), Oct-CA(1–7)M(2–9) and CA(1–7)M 16 mg/kg 4 h post-infection with a single dose of each peptide (i.p.; diluted in 500 µL of saline)	↓ Bacterial load in peritoneal fluid (all peptides)	López-Rojas et al., 2011 [27]
Neutropenic BALB/c mice (female) ^a^ Open wound 10^7^ CFU/mouse pipetted into the wound (a 6 mm disposable skin biopsy Punch) at a volume of 0.01 mL in saline	K11 4 µg/dose/mouse 4 h post-infection followed by application every 24 h for 21 days (topical; hydrogel)	↓ Body weight loss ↑ Bacterial clearance in wounds ↑ Healing ↓ Levels of malondialdehyde in wounds ↑ Levels of catalase in wounds	Rishi et al., 2018 [28]
Non-neutropenic BALB/c mice (male) Peritoneal infection 10^8^ CFU/mL dissolved in 500 µL PBS i.p.	Hylin a1-11K and Hylin a1-15K 1 and 2 mg/kg 1 h post-infection with a single dose of each peptide (i.p.; diluted in 500 µL of PBS)	↓ Lethality ↓ Inflammation in the spleen, lung, liver, and kidney	Park et al., 2022 [29]
Non-neutropenic BALB/c mice (male) Burn wound 10^5^ CFU/mouse (20 µL in Mueller–Hinton broth) was inoculated in the burn wound area (using a metallic plate of pure iron with a surface area of 0.785 cm^2^ heated in water at 100 °C added on the skin by 10–20 s)	Melittin 8, 16, and 32 μg/mL 1 h post-infection with a single dose of each concentration (topical; diluted in water). After the addition of peptide, it was incubated on the infected area for 2 h	↓ Bacterial load in burn wound (all concentrations)	Pashaei et al., 2019 [30]
Non-neutropenic BALB/c mice Peritoneal infection 10^7^ CFU/mL dissolved in 500 µL saline i.p.	Mastoparan and Chitosan–mastoparan nanoconstruct Not shown 1 h post-infection with a single dose (i.p.)	↓ Bacterial load in blood (chitosan–mastoparan nanoconstruct) - No decrease in the bacterial load in blood (mastoparan)	Hassan et al., 2021 [31]
Non-neutropenic BALB/c mice (female) Pulmonary infection 10^8^ CFU/mice dissolved in 20 µL saline i.n.	LyeTx I-b and LyeTx I-b pegylated 0.5, 1, and 2 mg/kg 2 h post-infection with a single dose of each concentration (i.v.; 100 µL of saline)	↓ Bacterial load in the lung (LyeTx I-b pegylated) - No decrease in the bacterial load in the lung (LyeTx I-b)	Brito et al., 2021 [23]
Non-neutropenic BALB/c mice (female) Pulmonary infection 10^8^ CFU/mice dissolved in 20 µL saline i.n.	LyeTx I mnΔK 1, 5, and 2 mg/kg 2 h post-infection with a single dose of each concentration (i.n.; 20 µL of saline)	↓ Bacterial load in the lung	Lima et al., 2021 [32]

i.p.: intraperitoneally; i.v. intravenous; i.n.: intranasal; CFU: Colony-forming units. ^a^ Neutropenia protocol: Cyclophosphamide at 150 mg/kg body weight (200 µL) 4 days and 1 day prior to infection.

## Data Availability

Not applicable.

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
