# Peer review of "Therapeutic Prospection of Animal Venoms-Derived Antimicrobial Peptides against Infections by Multidrug-Resistant Acinetobacter baumannii: A Systematic Review of Pre-Clinical Studies"

_toxins, 2023, doi:10.3390/toxins15040268_

Round 1
Reviewer 1 Report
Reference: “Therapeutic prospection of animal venoms-derived antimicro- 2 bial peptides against infections by multidrug-resistant Acineto- 3 bacter baumannii: A systematic review of pre-clinical studies” submitted to Toxins 2023.
General comments: In the manuscript presented, the authors are reporting a review on the possible actions of toxins from venomous animals on multidrug-resistant Acinetobacter baumannii (MDR-Ab), a bacterium that is a public health problem described worldwide, due to the low availability of antibiotics and other functional antimicrobials for the treatment of infections. Throughout the text, a systematic review and a Meta-analysis of eight scientific publications are shown that describe eleven peptides obtained from animal venoms with antimicrobial activities on multidrug-resistant Acinetobacter baumannii. The authors, based on published data, describe the potential of different toxins to be used as sources of antimicrobials in the multidrug-resistant treatment of Acinetobacter baumannii. After careful reading of the text, it is my opinion that it can be published by TOXINS, since the subject is within the Journal scope and the text produced is well written, showing the group's knowledge on the subject. However, here are suggestions that can make a revised text more attractive to readers in the area.
Specific Comments
1- In the line 32 …. to become resistant to several antimicrobials [2,3]. Although the authors cite two very interesting references, I would write in this sentence the names of some antibiotics currently used in the clinic and which are inhibited by enzymes and molecules produced by this bacterium. This makes the text more attractive to clinicians and researchers in the area.
2- Between lines 52 to 54 …. These potential therapeutic agents have shown promising activity against several species of bacteria of medical interest, including strains resistant to conventional antimicrobials [17]. Once again, although the authors cite a pertinent reference, it would be interesting, and this would make the text more attractive, to mention the names of the bacteria that are targets of AMPs.
3- Lines 55 to 57 …. or inhibition of intracellular targets, and are also capable of modulating the antibacterial immune response [18]. This phrase could be further explored and detailed, where the authors could indicate which intracellular structures can be inhibited by AMPs in addition to the types of immune response.
4- Line 60 … when combined with commercial antimicrobials [13-18]. I would add the word bactericidal antimicrobials to differentiate from bacteriostatic antimicrobials, which seems to me not to be the case for recommended combinations with AMPs, since AMPs are bactericidal.
5- Lines 65 and 66 …. several animals, such as ants, frogs, wasps, bees, spiders, scorpions, molluscs, snakes, and myriapods [20]. Again although a reference is indicated, I would cite the genera and species where AMP productions have been described.
6- Lines 66 to 69 … These molecules are often used as prototypes for developing analogues with better therapeutic properties, such as greater pharmacological potency, lower toxicity, and a pharmacokinetic profile more appropriate [21,22]. As written in this sentence, it seems that there are already analogous molecules developed using AMPs as templates and already used in the clinic. If this is true then indicate the names of these synthetic analogues and more details of their pharmacological behaviors.
7- Lines 78 and 79 …. selected databases (100 in Scopus, 98 in PubMed/MEDLINE, 32 in ScienceDirect, 22 in Web of Science, and 6 in Biblioteca Virtual em Saúde In this sentence, without a doubt, the authors should indicate the electronic addresses of each database studied, this would leave the text in a more scientific language and provide correct access to readers interested in the subject.
8- About the text described between lines 77 to 89, and figure 1, I find it interesting that the authors describe in the text what were the main criteria (key words) used to arrive at eight publications of interest in this review. Although the authors indicate reference 32, the text as it is presented in this first version of the manuscript, I cannot identify clear criteria to reach the inclusions or exclusions of publications. In a corrected version I suggest that authors make a brief description of the conditions used to exclude or include publications of interest, especially in the identification and screening criteria!
9- Such as comments could also be incorporated in legends of figure1.
10- In table 1, described between lines 135 to 157, in the last column, where the references of the data used in the table are cited, the authors could add the number of references in parentheses, as they are organized throughout the text.
11- Regarding table 2, described between lines 202 to 203, I suggest that the authors change the words molecular weight to molecular mass. Molecules have molecular mass, not molecular weight. There was never talk of weight spectrometry, but instead mass spectrometry!
12- I found it strange that although there are variations in the numbers of basic amino acid residues in the different peptides, the authors describe that they all have a pI of 14? In Table 2. Would I like any comments from the authors on this?
13- I also believe that data on water solubility of different peptides would be interesting.
14- In the version of manuscript that I received, the words analogues appear separately in column 1, of Table 2 (ana-logues). The authors could reformat and correct this in the revised version.
15- Also regarding Table 2, if the data shown are not studies of the authors of this manuscript, then cite references as they appear in this text!
16- In Table 3, lines 231 to 256, the authors should place the cited references in the last column in numbers, as they appear in the text.
17- Also with regard to Table 3, I would just leave a black circle preceding each model studied. As shown it gets confusing. In my opinion, you don't need to put a black circle per line, as they are data from the same study and model. Only one black circle per different model studied.
18- In my oppinion sub chapers 2.4.4. Bacterial load, 2.4.5. Physical results, 2.4.6. Inflammatory and oxidative responses could be incorporated in the text before Table 3, as they discuss aspects shown in the Table 3.
19- In the Discussion chapter, between lines 325 to 327 the authors wrote …. AMPs can be derived from the venom of numerous animals, and in this review, arthropod venom was highlighted as the main source of active compounds against MDRAb. This sentence seemed confusing to me! Perhaps better to be rewritten. As it stands, it appears that arthropods are the main source of AMPs, and this may not be the case. Only the authors decided to study arthropods in this text, but certainly other animals, fungi and plants also have AMPs that could be useful in the clinic.
20- The authors discuss in the Discussion chapter, between lines 338 to 354, the participation of carbapenemases as important molecules in the mechanism of resistance of microorganisms to antibiotics. However, I did not see direct relationships between treatments with AMPs and inhibition of these enzymes. Is there any direct relationship? This could be discussed in the revised text!
21- I missed in the text description of AMPs activities on the immune system. Could these peptides stimulate antibody production or activate immune system cells? Generating an immune response capable of inhibiting future uses of these products?
22- Also did I miss data on the concentrations of the peptides in the blood after the various administrations and half-life in the bloodstream? In addition to stability in the bloodstream, which are important data for thinking about therapies in the clinical routine.
23- I missed descriptions of the possible side effects caused by the administration of the studied peptides. Surely they exist and need to be verified.
24- Discussion chapter lines 400 to 402 thje authors wrote ….. The MDR-Ab infection models used mainly inbred mice from the BALB/c strain. The BALB/c mouse is among the most widely used models in biomedical research and is particularly employed in immunology and infectious disease studies. All the more reason to have data on the immunological characteristics of AMPs.
25- I missed data on the actions of treatments with AMPs on functional systems and organs. Biochemical parameters dosages proving the absence of alterations in renal, hepatic, hematological, cerebral, cardiac functions, among others. These are essential data to ensure the safety of future clinical applications and should be among the first tests to be carried out, in addition to antibacterial efficiency. Is no data available in the literature?
26- The Conclusion of the text described between lines 477 to 491 is repetitive and could be excluded, since it has already been discussed throughout the text, mainly in the discussion.
27- Finally, the authors should read the entire text and define in the revised version the abbreviations that appear throughout the text, but are not defined the first time they are cited in the text.
Author Response
Reviewer #1
General comments: In the manuscript presented, the authors are reporting a review on the possible actions of toxins from venomous animals on multidrug-resistant Acinetobacter baumannii (MDR-Ab), a bacterium that is a public health problem described worldwide, due to the low availability of antibiotics and other functional antimicrobials for the treatment of infections. Throughout the text, a systematic review and a Meta-analysis of eight scientific publications are shown that describe eleven peptides obtained from animal venoms with antimicrobial activities on multidrug-resistant Acinetobacter baumannii. The authors, based on published data, describe the potential of different toxins to be used as sources of antimicrobials in the multidrug-resistant treatment of Acinetobacter baumannii. After careful reading of the text, it is my opinion that it can be published by TOXINS, since the subject is within the Journal scope and the text produced is well written, showing the group's knowledge on the subject. However, here are suggestions that can make a revised text more attractive to readers in the area.
Specific Comments
1 - In the line 32 …. to become resistant to several antimicrobials [2,3]. Although the authors cite two very interesting references, I would write in this sentence the names of some antibiotics currently used in the clinic and which are inhibited by enzymes and molecules produced by this bacterium. This makes the text more attractive to clinicians and researchers in the area.
Answer: The article has been corrected as suggested by the reviewer.
2- Between lines 52 to 54 …. These potential therapeutic agents have shown promising activity against several species of bacteria of medical interest, including strains resistant to conventional antimicrobials [17]. Once again, although the authors cite a pertinent reference, it would be interesting, and this would make the text more attractive, to mention the names of the bacteria that are targets of AMPs.
Answer: The article has been corrected as suggested by the reviewer.
3- Lines 55 to 57 …. or inhibition of intracellular targets, and are also capable of modulating the antibacterial immune response [18]. This phrase could be further explored and detailed, where the authors could indicate which intracellular structures can be inhibited by AMPs in addition to the types of immune response.
Answer: The article has been corrected as suggested by the reviewer.
4- Line 60 … when combined with commercial antimicrobials [13-18]. I would add the word bactericidal antimicrobials to differentiate from bacteriostatic antimicrobials, which seems to me not to be the case for recommended combinations with AMPs, since AMPs are bactericidal.
Answer: The article has been corrected as suggested by the reviewer.
5- Lines 65 and 66 …. several animals, such as ants, frogs, wasps, bees, spiders, scorpions, molluscs, snakes, and myriapods [20]. Again although a reference is indicated, I would cite the genera and species where AMP productions have been described.
Answer: AMPs can be isolated from an infinitely large number of animal species. As this is the introduction of the article, which according to the rules of the journal itself should contain the most relevant information of the study, we decided, in this case, to add in a more general way to avoid long lists of species already in the introduction. Species identification was reserved for results. We appreciate the suggestion by the editor, but we would like to maintain the following structure to adhere to the journal's guidelines and avoid overly large descriptions in this section.
6- Lines 66 to 69 … These molecules are often used as prototypes for developing analogues with better therapeutic properties, such as greater pharmacological potency, lower toxicity, and a pharmacokinetic profile more appropriate [21,22]. As written in this sentence, it seems that there are already analogous molecules developed using AMPs as templates and already used in the clinic. If this is true then indicate the names of these synthetic analogues and more details of their pharmacological behaviors.
Answer: The article has been corrected as suggested by the reviewer.
7- Lines 78 and 79 …. selected databases (100 in Scopus, 98 in PubMed/MEDLINE, 32 in ScienceDirect, 22 in Web of Science, and 6 in Biblioteca Virtual em Saúde In this sentence, without a doubt, the authors should indicate the electronic addresses of each database studied, this would leave the text in a more scientific language and provide correct access to readers interested in the subject.
Answer: The article has been corrected as suggested by the reviewer.
8- About the text described between lines 77 to 89, and figure 1, I find it interesting that the authors describe in the text what were the main criteria (key words) used to arrive at eight publications of interest in this review. Although the authors indicate reference 32, the text as it is presented in this first version of the manuscript, I cannot identify clear criteria to reach the inclusions or exclusions of publications. In a corrected version I suggest that authors make a brief description of the conditions used to exclude or include publications of interest, especially in the identification and screening criteria!
Answer: Thanks for your suggestions. All inclusion and exclusion criteria for articles are detailed in Figure 1 and in the Methods section, as suggested by PRISMA 2021.
9- Such as comments could also be incorporated in legends of figure1.
Answer: Thanks for your suggestion. Figure 1 lists the reasons that led to the exclusion of articles after reading the full text. Regarding the reasons for exclusion in the screening of title, abstract and keywords, the PRISMA 2021 guide is clear in indicating that the reasons that lead to inclusion and exclusion at this stage do not need to be added in the figure or even in the text. Therefore, we decided to be faithful to the PRISMA 2021 guide to ensure that our study complies with international standards for systematic review.
10- In table 1, described between lines 135 to 157, in the last column, where the references of the data used in the table are cited, the authors could add the number of references in parentheses, as they are organized throughout the text.
Answer: The article has been corrected as suggested by the reviewer.
11- Regarding table 2, described between lines 202 to 203, I suggest that the authors change the words molecular massto molecular mass. Molecules have molecular mass, not molecular weight. There was never talk of weight spectrometry, but instead mass spectrometry!
Answer: The article has been corrected as suggested by the reviewer.
12- I found it strange that although there are variations in the numbers of basic amino acid residues in the different peptides, the authors describe that they all have a pI of 14? In Table 2. Would I like any comments from the authors on this?
Answer: All active peptides have a considerable number of basic residues with no acidic residues available. Peptides with this configuration have very high isoelectric points because they need to deprotonate all of their basic groups to reach a neutral state. Thus, the high isoelectric point of all compounds included is justified.
13- I also believe that data on water solubility of different peptides would be interesting.
Answer: Water solubility data has been added in Table 2.
14- In the version of manuscript that I received, the words analogues appear separately in column 1, of Table 2 (ana-logues). The authors could reformat and correct this in the revised version.
Answer: The Table 2 has been corrected as suggested by the reviewer.
15- Also regarding Table 2, if the data shown are not studies of the authors of this manuscript, then cite references as they appear in this text!
Answer: The Table 2 has been corrected as suggested by the reviewer.
16- In Table 3, lines 231 to 256, the authors should place the cited references in the last column in numbers, as they appear in the text.
Answer: The Table 3 has been corrected as suggested by the reviewer.
17- Also with regard to Table 3, I would just leave a black circle preceding each model studied. As shown it gets confusing. In my opinion, you don't need to put a black circle per line, as they are data from the same study and model. Only one black circle per different model studied.
Answer: The Table 3 has been corrected as suggested by the reviewer.
18- In my oppinion sub chapers 2.4.4. Bacterial load, 2.4.5. Physical results, 2.4.6. Inflammatory and oxidative responses could be incorporated in the text before Table 3, as they discuss aspects shown in the Table 3.
Answer: The article has been corrected as suggested by the reviewer.
19- In the Discussion chapter, between lines 325 to 327 the authors wrote …. AMPs can be derived from the venom of numerous animals, and in this review, arthropod venom was highlighted as the main source of active compounds against MDRAb. This sentence seemed confusing to me! Perhaps better to be rewritten. As it stands, it appears that arthropods are the main source of AMPs, and this may not be the case. Only the authors decided to study arthropods in this text, but certainly other animals, fungi and plants also have AMPs that could be useful in the clinic.
Answer: The article has been corrected as suggested by the reviewer.
20- The authors discuss in the Discussion chapter, between lines 338 to 354, the participation of carbapenemases as important molecules in the mechanism of resistance of microorganisms to antibiotics. However, I did not see direct relationships between treatments with AMPs and inhibition of these enzymes. Is there any direct relationship? This could be discussed in the revised text!
Answer: Thanks for the comment. AMPs have good activity against carbapenem-resistant Acinetobacter baumannii strains. In this paragraph, we seek to highlight the epidemiological relevance of this microorganism, with the aim of showing the importance that AMPs can have in combating these microorganisms. However, there is no evidence that AMPs inhibit the synthesis or activity of carbapenemases such as mentioned by reviewer.
21- I missed in the text description of AMPs activities on the immune system. Could these peptides stimulate antibody production or activate immune system cells? Generating an immune response capable of inhibiting future uses of these products?
Answer: The effect of AMPs on the immune system has been added to the text in the introduction section, as suggested by the reviewer (Lines 62-64).
22- Also did I miss data on the concentrations of the peptides in the blood after the various administrations and half-life in the bloodstream? In addition to stability in the bloodstream, which are important data for thinking about therapies in the clinical routine.
Answer: Thanks for your comment. As described in the “study limitations” section, none of the included articles evaluated the stability or pharmacokinetics of the studied peptides. Therefore, through a critical stance, we highlight that future studies need to address these issues. This was pointed out in the limitations of the study, as well as in the concluding section. Unfortunately, we were unable to discuss these data because the authors neglect them. The same goes for the safety profile of these compounds, which has been very little explored as highlighted in our review.
23- I missed descriptions of the possible side effects caused by the administration of the studied peptides. Surely they exist and need to be verified.
Answer: As described in the question above, the authors neglect safety data for the compounds. Therefore, we added this question to the limitations of the study.
24- Discussion chapter lines 400 to 402 thje authors wrote ….. The MDR-Ab infection models used mainly inbred mice from the BALB/c strain. The BALB/c mouse is among the most widely used models in biomedical research and is particularly employed in immunology and infectious disease studies. All the more reason to have data on the immunological characteristics of AMPs.
Answer: As mentioned in the previous questions addressed by the reviewer, these data have been neglected in different studies. Therefore, we were unable to explore the discussion in this sense, but we highlight the importance of studying these aspects in future studies in the study limitations section.
25- I missed data on the actions of treatments with AMPs on functional systems and organs. Biochemical parameters dosages proving the absence of alterations in renal, hepatic, hematological, cerebral, cardiac functions, among others. These are essential data to ensure the safety of future clinical applications and should be among the first tests to be carried out, in addition to antibacterial efficiency. Is no data available in the literature?
Answer: The reviewer, as well as our group, noticed the clear neglect of pharmacokinetics and safety data in studies conducted with AMPs. During the conduct of the review, this lack of essential information for pharmaceutical development caught our attention and was the result of great concern about how studies in the field of pharmaceutical development are being designed. There is a lack of important evidence in the evaluation of the real therapeutic potential of these compounds. Due to the absence of these data, a more comprehensive discussion of this topic cannot be conducted, but we highlight all these concerns in the study limitations section.
26- The Conclusion of the text described between lines 477 to 491 is repetitive and could be excluded, since it has already been discussed throughout the text, mainly in the discussion.
Answer: The article has been corrected as suggested by the reviewer.
27- Finally, the authors should read the entire text and define in the revised version the abbreviations that appear throughout the text, but are not defined the first time they are cited in the text.
Answer: The article has been corrected as suggested by the reviewer.
Reviewer 2 Report
Minor revision:
Page 3 line 112: Recheck this word “the month”, should be “the moth”
Page 3 line 118: Delete “animal venom” and replace by “honey bee and wasp venom”
Page 4 line 167: Recheck the decimal number
Page 5 line 171: Recheck the word “mastoparam”
Page 5 line 184: In section 2.4.1 Infection models, the authors presented only murine model as also mentioned in the key words, “rodent” in Page 15 line 518 .Could you add more in vivo models such as the nematode Caenorhabditis elegans and the Galleria mellonella models in the Discussion section?
Page 10 line 289-91: Rewrite the sentence to the positive result after AMP administration in the infected murine model.
Page 15 References section: Scientific name should be italic letter

Author Response
Reviewer #2
1 - Page 3 line 112: Recheck this word “the month”, should be “the moth”
Answer: The article has been corrected as suggested by the reviewer.
2 - Page 3 line 118: Delete “animal venom” and replace by “honey bee and wasp venom”
Answer: The article has been corrected as suggested by the reviewer.
3 - Page 4 line 167: Recheck the decimal number
Answer: The article has been corrected as suggested by the reviewer.
4 - Page 5 line 171: Recheck the word “mastoparam”
Answer: The article has been corrected as suggested by the reviewer.
5 - Page 5 line 184: In section 2.4.1 Infection models, the authors presented only murine model as also mentioned in the key words, “rodent” in Page 15 line 518 .Could you add more in vivo models such as the nematode Caenorhabditis elegans and the Galleria mellonella models in the Discussion section?
Answer: Thanks for the comment. The objective of this review was to synthesize only the in vivo antibacterial activity data that used vertebrate models. In fact, the use of invertebrates was employed as an exclusion factor is this review. But we appreciate the reviewer's suggestion.
6 - Page 10 line 289-91: Rewrite the sentence to the positive result after AMP administration in the infected murine model.
Answer: The article has been corrected as suggested by the reviewer.
7 - Page 15 References section: Scientific name should be italic letter
Answer: The article has been corrected as suggested by the reviewer.
Reviewer 3 Report
The review summarized the latest studies that showed the potential application of venom-derived AMPs to combat MDR-Ab. The systematic review was conducted based on pertinent criteria, and in vitro properties and in vivo results of AMPs were presented. The manuscript holds substantial value as it focused on in vivo studies, compiling models and methods that were used to test the antimicrobial activity of AMPs against MDR-Ab. However, there are some major issues to be addressed throughout the manuscript to meet the standard for the review article.
l As mentioned in the main text, MDR-Ab is a major concern to public health, and more information must be described. For instance, the authors should elaborate on the distinctive characteristics of MDR-Ab compared with other pathogens as well as common Ab strains. Also, the major reasons for high risk, lethality, and occurrence of MDR-Ab infection should be included in the introduction.
l The review should include more detailed and comprehensive information on venom-derived AMPs to let readers understand the therapeutic potential and value of animal venom.
n How were the AMPs initially identified from the animal venom?
n Were there other cases where animal venom-derived AMPs showed anti-MDR-bacterial activity? If so, what are the similarities and differences among them?
n Did the selected AMPs show broad antimicrobial activity or were specific to MDR-Ab?
n In the case of synthesized AMPs, how were original templates first selected? What rationale was used to produce derivatives? Is there a functional difference?
l A brief summary or explanation of the biological mechanism of the AMP activity that was explained in each study can be helpful for the readers.
l Since biofilm formation contributes to drug resistance, anti-MDR-Ab biofilm activity and mechanism of peptides will support the therapeutic importance of the peptides. Is there in vivo data or at least in vitro data that can be included?
l The efficacy or therapeutic index of the selected peptide should be suggested and compared with the conventional antibiotics that was used in the original research.
l It will help the readers if the authors discuss the intercorrelation between the animal source and the identified AMPs based on the features of AMPs.
Some statistical data are out of date; they should be renewed. Also, reconsider the format of the manuscript.
Author Response
Reviewer #3
The review summarized the latest studies that showed the potential application of venom-derived AMPs to combat MDR-Ab. The systematic review was conducted based on pertinent criteria, and in vitro properties and in vivo results of AMPs were presented. The manuscript holds substantial value as it focused on in vivo studies, compiling models and methods that were used to test the antimicrobial activity of AMPs against MDR-Ab. However, there are some major issues to be addressed throughout the manuscript to meet the standard for the review article.
l - As mentioned in the main text, MDR-Ab is a major concern to public health, and more information must be described. For instance, the authors should elaborate on the distinctive characteristics of MDR-Ab compared with other pathogens as well as common Ab strains. Also, the major reasons for high risk, lethality, and occurrence of MDR-Ab infection should be included in the introduction.
Answer: The information suggested by the reviewer was added in the manuscript.
2 - The review should include more detailed and comprehensive information on venom-derived AMPs to let readers understand the therapeutic potential and value of animal venom.
Answer: The information suggested by the reviewer was added in the manuscript.
3 - How were the AMPs initially identified from the animal venom?
Answer:
4 - Were there other cases where animal venom-derived AMPs showed anti-MDR-bacterial activity? If so, what are the similarities and differences among them?
Answer: There are different ways to identify AMPs from the toxin of animals, among them tests of fractions of the venom, synthesis of compounds identified by spectrometric techniques, testing of the venom after treatment that eliminates major compounds, among other various ways. We believe this topic is too broad to be covered in a review focused on antibacterial activity like this one, but we appreciate reviewer scores.
5 - Did the selected AMPs show broad antimicrobial activity or were specific to MDR-Ab?
Answer: Unfortunately, due to the focus of the research question, we only included studies that evaluated the activity of AMPs against MDR-Ab, so we do not know the antimicrobial spectrum of these compounds.
6 - In the case of synthesized AMPs, how were original templates first selected? What rationale was used to produce derivatives? Is there a functional difference?
Answer: Thanks to the reviewer for the comment. The chemical origin of the compounds is described in table 1, and some data referring to this process were discussed in the manuscript.
7 - A brief summary or explanation of the biological mechanism of the AMP activity that was explained in each study can be helpful for the readers.
Answer: The general mechanism of action of AMPs was described in the introduction section as suggested by the reviewer.
8 - Since biofilm formation contributes to drug resistance, anti-MDR-Ab biofilm activity and mechanism of peptides will support the therapeutic importance of the peptides. Is there in vivo data or at least in vitro data that can be included?
Answer: Thanks to the reviewer for the comment. All of the studies we included in this review evaluated antibacterial activity against planktonic cells.
9 - The efficacy or therapeutic index of the selected peptide should be suggested and compared with the conventional antibiotics that was used in the original research.
Answer: Only one study evaluated the safety of the AMP used. This neglect of toxicity studies was highlighted in the limitations section of our manuscript.
10 - It will help the readers if the authors discuss the intercorrelation between the animal source and the identified AMPs based on the features of AMPs.
Answer: The source of the main AMPs included in this study was reported in the Discussion section.
11 - Some statistical data are out of date; they should be renewed. Also, reconsider the format of the manuscript.
Answer: Some references and data have been updated as suggested by the reviewer.
Round 2
Reviewer 1 Report
Dear Editors and Authors.
After careful reading of the revised text it is my opinion that the authors have made a good revision, with good changes in the text. They answered the reviewer's questions and produced a revised text that was better than the original form. Therefore, in my opinion this version can be accepted for publication in TOXINS. The subject falls within the scope of the Journal, the text was well presented, coherent and has scientific appeal. Best Regards. Congratulations.
Reviewer 3 Report
Accept in present form